

# Development of a cryptocurrency price prediction model: leveraging GRU and LSTM for Bitcoin, Litecoin and Ethereum

Ramneet Kaur[1], Mudita Uppal[1], Deepali Gupta[1], Sapna Juneja[2], Syed Yasser Arafat[3], Junaid Rashid[4], Jungeun Kim[5] and Roobaea Alroobaea[6]

[1] Chitkara University Institute of Engineering and Technology, Chitkara University, Punjab, India
[2] KIET Group of Institutions, Ghaziabad, India
[3] Mirpur University of Science and Technology, Mirpur, Pakistan
[4] Department of Artificial Intelligence and Data Science, Sejong University, Seoul, Republic of South Korea
[5] Department of Computer Science and Engineering, Inha University, Incheon, Republic of South Korea
[6] Department of Computer Science, College of Computers and Information Technology, Taif University, Taif, Saudi Arabia

Corresponding authors
Junaid Rashid,
junaidrashid062@gmail.com
Jungeun Kim, jekim@inha.ac.kr

## ABSTRACT

Cryptocurrency represents a form of asset that has arisen from the progress of financial technology, presenting significant prospects for scholarly investigations. The ability to anticipate cryptocurrency prices with extreme accuracy is very desirable to researchers and investors. However, time-series data presents significant challenges due to the nonlinear nature of the cryptocurrency market, complicating precise price predictions. Several studies have explored cryptocurrency price prediction using various deep learning (DL) algorithms. Three leading cryptocurrencies, determined by market capitalization, Ethereum (ETH), Bitcoin (BTC), and Litecoin (LTC), are examined for exchange rate predictions in this study. Two categories of recurrent neural networks (RNNs), specifically long short-term memory (LSTM) and gated recurrent unit (GRU), are employed. Four performance metrics are selected to evaluate the prediction accuracy namely mean squared error (MSE), mean absolute error (MAE), mean absolute percentage error (MAPE), and root mean squared error (RMSE) for three cryptocurrencies which demonstrates that GRU model outperforms LSTM. The GRU model was implemented as a two-layer deep learning network, optimized using the Adam optimizer with a dropout rate of 0.2 to prevent overfitting. The model was trained using normalized historical price data sourced from CryptoDataDownload, with an 80:20 train-test split. In this work, GRU qualifies as the best algorithm for developing a cryptocurrency price prediction model. MAPE values for BTC, LTC and ETH are 0.03540, 0.08703 and 0.04415, respectively, which indicate that GRU offers the most accurate forecasts as compared to LSTM. These prediction models are valuable for traders and investors, offering accurate cryptocurrency price predictions. Future studies should also consider additional variables, such as social media trends and trade volumes that may impact cryptocurrency pricing.

## INTRODUCTION

Cryptocurrency is any virtual or digital currency that is used in financial systems to prevent counterfeiting, with encryption serving as the primary form of protection (*Mukhopadhyay et al., 2016*; *Rose, 2015*). It is different from traditional currencies as it is not controlled by banks or central authorities and because it is decentralized, it may be converted using some cryptographic techniques (*Brenig, Accorsi & Müller, 2015*). Another characteristic that made it feasible is by using a much known technology called blockchain which stores data that is hard to hack, change or deceive (*Eyal, 2017*). It is hard to predict when cryptocurrencies will become extensively used in international markets because they are still in their infancy (*Mäntymäki, Wirén & Najmul Islam, 2020*). The widely used cryptocurrency, Bitcoin (BTC) first appeared in 2009 as a trailblazer in the field of blockchain technology. For more than two years, it was the only digital money in the blockchain space. The development of cryptocurrencies is still ongoing and it is still unclear how widely they will be accepted and integrated in the future. There are already over 5,000 digital currencies in circulation in the cryptocurrency market and there are an estimated 5.8 million active individuals involved in it (*Jang & Lee, 2017*). Because of the unusual combination of monetary units and encryption technology, BTC has been attracted from a variety of industries, including economics, computer science and cryptography. This multidisciplinary effect and expanding relevance of cryptocurrencies in modern study and conversation are highlighted by combination of different fields (*Saad et al., 2019*).

The core technology of the Bitcoin cryptocurrency system, known as blockchain, is essential in improving security and privacy different areas, most preferably in Internet of Things (IoT) environment. Blockchain acts as a digital ledger that distributes the recording of transactions throughout a network of computer systems in a decentralized manner. The potential of blockchain technology to transform trust mechanisms and data integrity in various industries is evident from its use outside of the cryptocurrency space. This highlights the significance of blockchain in influencing the direction of digital infrastructure and innovation in the future (*Gautam, Sharma & Kumar, 2020*; *Adams, Kewell & Parry, 2018*). Blocks and transactions are the two main components of a blockchain. Blocks operate as data collections that record transactions together with related information like sequence and timestamp, whereas transactions represent participant activities. Blockchain also includes a signaling system called BloSS, a cooperative DDoS defense system that operates across several domains using blockchain technology. In this system, each autonomous system works together as a member of the defensive alliance. This paradigm emphasizes how blockchain may cooperate with others to strengthen security protocols in a variety of contexts (*Killer, Rodrigues & Stiller, 2019*). According to *Gandal & Halaburda (2014)*, there are two factors that influence how

networks affect competition in the emerging cryptocurrency industry in terms of exchange rates over time:

(1) competition between various currencies.

(2) competition among exchanges.

Despite the fact that there are hundreds of cryptocurrencies present in the market, but Bitcoin is the most known as it is a tenacious rival and has not yet broken free from the cryptocurrency battle. It has thus emerged as the leading cryptocurrency. A competition was observed among cryptocurrencies as "healthy competition," supporting continued breakthroughs in technology and security innovations (*Iwamura, Kitamura & Matsumoto, 2014*). However, *Kyriazis (2019)* shows that BTC and conventional national currencies demonstrate volatility shock transmission and that the phenomena are largely unaffected by uncertainty in economic policy. These revelations highlight the complex relationship that exists between cryptocurrencies and traditional monetary systems, highlighting the role that economic stability and technical robustness will help in determining the direction that digital currencies will drive in the future. The authors explored how big data and cryptocurrencies interact, presenting the Bitcoin market as a seductive setting for financial speculation (*Hassani, Huang & Silva, 2018*). It also contributed to the growth of dishonest practices made possible by social media platforms. This investigation clarifies the intricate relationships between digital currencies, data analytics and the difficulties presented by dishonest activities in the cryptocurrency space. Many people have made significant gains from speculative investments in digital marketplaces. But all investments include some danger, which is why some investors—especially those who are willing to take on more risk—tend to invest in cryptocurrencies. As these markets are volatile, market analysts and speculators rely largely on predictive approaches to navigate and profit from them (*Nizzoli et al., 2020*; *Rebane et al., 2018*).

Utilizing machine learning (ML) and deep learning (DL) algorithms is of moderate interest, given the variations in predicting effectiveness among various cryptocurrencies. These cutting-edge technologies provide exciting opportunities for improving prediction performance in the cryptocurrency space, adjusting to the volatility that are specific to each virtual asset (*Rehman & Apergis, 2019*). Cryptocurrencies that are less volatile than those that are more volatile are more predictable. Furthermore, different data sets have different levels of efficiency for various ML algorithms and *via* using these approaches, it makes prediction much more complex. This emphasizes the complex issues surrounding Bitcoin forecasting and how crucial it is to comprehend how volatility levels and prediction techniques interact (*Liew et al., 2019*). Even with the use of cryptocurrencies for a variety of transactions throughout the world, still there is disagreement about the exact meaning of a cryptocurrency and its legal status. It also highlights how difficult is to categorize and regulate digital currencies, which reflects how legal frameworks are always changing to keep up with the cutting-edge nature of Bitcoin technology (*Dyntu & Dykyi, 2018*). Moreover, the previously indicated situation amplifies the challenges faced in criminal inquiries concerning cryptocurrency-driven money laundering. Thus, law enforcement organizations encounter difficulties in locating criminals and providing evidence of illegal activity, which makes it difficult to determine guilt and bring criminal charges against

offenders. This emphasizes how urgently improved regulatory controls and investigative resources are needed to properly handle illegal activity in the Bitcoin space (*Kethineni & Cao, 2020*). Focus on Bitcoin price is similar to focus on stock price, but one important difference is that cryptographic forms of money are not exposed to similar risk factors that influence stock price changes. Furthermore, most of digital forms of money are to a great extent unaffected by traditional macroeconomic pointers, for example, currency rates, item costs and larger economic issues affecting conventional assets. The distinct dynamics and elements influencing digital asset pricing in the Bitcoin market are highlighted in this article (*Liu & Tsyvinski, 2021*).

In the 2017 cryptocurrency market boom, some governments worldwide started to standardize and regulate digital currency. Due to the safety of blockchain technology, Bitcoin is now used by more people in a more comfortable manner (*Valdeolmillos et al., 2020*). Blockchain technology provides a strong security foundation, but those who use cryptocurrencies for illegal purposes must also be taken into consideration when accounting for the legality of cryptocurrencies. As a result, there are continuous discussions and examinations around the legality of cryptocurrencies. They emphasize how complex technological innovation, legal frameworks and moral issues interact in digital currency. The viewpoints and characteristics of cryptocurrencies in respect of monetary elements, legal and economic issues are discussed by the authors (*Yuneline, 2019*). From an economic perspective, Bitcoin does not meet the requirements of a currency on the basis of the viewpoint and features of traditional currency. In digital markets, BTC is a well-known cryptocurrency among many. Because cryptocurrencies have significant interrelationships, smaller cryptocurrencies can cause shocks that can harm larger cryptocurrencies. Based on research (*Huynh et al., 2020*), gold acts as a stand-alone currency and can be a useful asset for hedging against unanticipated changes in the price of cryptocurrencies. This highlights the possibility that conventional assets, like gold, might operate as stabilizing elements in the ever-changing world of virtual currencies providing opportunities for risk mitigation and diversification tactics.

The dynamic and inherently volatile nature of cryptocurrencies makes price forecasting particularly challenging. This study focuses on analyzing three prominent cryptocurrencies among the many worldwide. By delving into the complexities of these specific cryptocurrencies, this research aims to advance the understanding of factors driving price fluctuations in the cryptocurrency market. Therefore, the following objectives are established in this study, utilizing machine learning and deep learning algorithms that can identify latent patterns in data to achieve more precise predictions. The key contributions of this research are as follows:

- This article presents a comprehensive analysis of deep learning models used to predict the prices of three cryptocurrencies named Bitcoin, Ethereum and Litecoin.
- Two categories of recurrent neural networks, namely long short-term memory (LSTM) and gated recurrent unit (GRU), are utilized for cryptocurrency price prediction. The

study leverages these DL algorithms to enhance prediction accuracy and handle the inherent nonlinearities in time-series data.

- Experimental results represent that GRU model outperforms LSTM model in predicting cryptocurrency prices for BTC, LTC, and ETH, as indicated by lower MAPE values, thereby establishing GRU as the more accurate prediction algorithm.

This section gives an overview of cryptocurrency and rest of the article is organized as follows: "Related Work" discusses the literature review and past studies in specific area. "Materials and Methods" represents the statistical analysis of the data as well as the outcomes. "Results" explains the dataset used and demonstrates experimental outcomes. The model suggested in this research is compared to previous studies mentioned in literature, whereas "Discussion" summarizes the article's general findings.

## RELATED WORK

Deep learning is a powerful subfield of AI that can predict future events *via* the analysis of past data. Its advantage over conventional forecasting techniques is well-established, since it regularly produces findings that are almost exactly like real results. Additionally, machine learning's ability to improve outcome precision is apparent, that is why it is a favored option in a variety of fields. Moreover, because of its capacity to recognize intricate patterns and adjust to changing datasets, ML and DL are becoming a crucial component of decision-making processes in a variety of sectors (*Hitam & Ismail, 2018*).

The author's emphasized benefits of investing in cryptocurrencies using ML algorithms like neural networks (NN), support vector machines (SVM), and random forests (RF) (*Andrianto & Diputra, 2017*). Firstly Bitcoin has shown less standard deviation which expand the adequacy of methodology. Secondly, cryptocurrencies enable investors to diversify different investment strategies by giving them access to a wide range of allocation options. A total of 5% to 20% allocation is the optimal amount for allocation to cryptocurrencies which depends upon investor preferences and risk tolerance. To enhance the performance of their holdings, investors tailor their portfolios as per their risk profiles by using this range. The focus of author's investigation the field of time series data forecasting is prediction of BTC value (*Derbentsev et al., 2021*). Two machine learning algorithms RF and stochastic gradient boosting machine (SGBM) are used in this article that explain how well ensemble strategy work in this particular area. The results proved that these calculations can change the cost of Bitcoin and open-ups various opportunities for ML in surveying and navigation of digital money. In this study, ML techniques proved useful and adaptable in forecasting financial decisions for better understanding in cryptocurrency market environment. Timely and well informed decisions lessen the risk associated with investigating cryptocurrency for effective decision making is also necessary. The authors proposed a hybrid cryptocurrency prediction system that uses GRU and LSTM models (*Patel et al., 2020*). The proposed algorithm predicts the trajectories of two cryptocurrencies namely LTC and Monera. The main aim of research is to improve understanding of market dynamics by integrating various neural network designs for intelligent decision making by the investors. The combination of LSTM and GRU model

maximize forecast accuracy and help investors to navigate the Bitcoin market with greater confidence and efficiency. The authors observe the complexity of Bitcoin returns which are minute-sampled as well as aggregated over 3 h to provide volatility in data (*Miura, Pichl & Kaizoji, 2019*). The authors used different ML techniques namely SVM, ANN, GRU, LSTM to protect value of cryptocurrency on the basis of past samples. Heterogeneous Auto-Regressive Realized Volatility (HARRV) model is contrasted with these prediction models after the optimization of the lag values.

The results proved that the proposed model accurately protects the prices which highlights the potential application in forecasting the cryptocurrency value. In another work the authors used SVM and linear regression methods to protect the Bitcoin prices (*Karasu et al., 2018*). The authors used a time series data set which contains Bitcoin closing values to construct prediction models. The past data of price is combined with machine learning methods to provide patterns of Bitcoin values in future. These two models make well informed decisions for forecasting of cryptocurrency in this digital asset market. The authors proposed article learning method with gradient mechanism for Bitcoin price prediction and proposed a regression approach on the basis of correlated attributes (*Saad et al., 2019*). The searches focused on the prices of Bitcoin, etherum and Ripple and how it change in the field of cryptocurrency analysis. The study emphasis on characteristics of these models by using AI frameworks like RNN or ANN. This research proved that LSTM is better at capturing short term fluctuations in Bitcoin values whereas ANN is best at capturing long term fluctuations (*Yiying & Yeze, 2019*). In this field, LSTM helps in making well informed decisions as it can extract information from the past data. This review article showed various results about day to day value of Bitcoin using high layered information (*Chen, Li & Sun, 2020*). A total of 66% is exhibited by both direct discriminant examination and strategic relapse. But for daily price prediction an advanced ML algorithm performs much better than conventional techniques. A total of 66% and 65.3% was achieved by statistical method and ML algorithm respectively. This shows that how ml methods improve the accuracy of forecasting and provide insights of market and how predictive analytics is evolving in cryptocurrency markets. Investors looking to move more precisely and confidently through the unpredictable and complicated world of BTC trading will find these insights to be of immeasurable value. The study explores the effectiveness of neural networks, support vector machine and random forests in forecasting cryptocurrency prices (*Valencia, Gómez-Espinosa & Valdés-Aguirre, 2019*).

The study emphasized how machine learning methods can be used in conjunction with sentiment analysis to anticipate market movements. It also demonstrates that Twitter data alone can provide predictive power, particularly when it comes to forecasting individual coins. Curiously, the outcomes showed that NN are the most reliable model among the models that were examined, beating SVM and RF. It highlights the value of cutting-edge neural network designs in recognizing the complex correlations contained in cryptocurrency market data. The most significant technique for forecasting BTC values in the stock market is LSTM model that used Yahoo Finance data (*Ferdiansyah et al., 2019*). This work developed a method for anticipating Bitcoin values above $12,600 USD just followed by prediction. The researchers focused on ML techniques because it is necessary

to make reliable technique for Bitcoin price prediction. The aim of this work is to increase the accuracy of forecasting Bitcoin prices by using LSTM model that provide stakeholders with necessary information of digital asset marketIt focus on innovation that shows a proactive approach to address the challenges associated with market forecasting and predict the dynamics of stock datasets, a complete method integrated linear and non-linear time-series components into a hybrid model framework. The amalgamation of CNN and Sequence-to-Sequence LSTM networks (Seq2Seq LSTMs) proved useful in the non-linear time series forecast. The data patterns depend on both short-term and long-term timeframes, which can be captured because of the dynamic modeling made feasible by this link (*Zhao & Chen, 2022*). This work offered a comprehensive understanding of market patterns that improve the adaptability of forecasts of stock market by utilizing the advantages of CNN and Seq2Seq LSTMs. This integrated method indicates a comprehension of the different nature of financial time series data and a concentrated effort to construct trustworthy forecasting algorithms that are capable of precisely and successfully navigating the complexity of stock market movements.

The currencies BTC and LTC, which are quickly gaining popularity for online transactions globally, are analysed using a multi-straight relapse model that prioritises cooperative perspectives. The R2 values for LTC and BTC were 44% and 59%, respectively, according to *Jain et al. (2018)*. In this investigation, two distinct LSTM models were employed: a regular LSTM and an LSTM coupled with an AR(2) model (*Wu et al., 2018*). This study showcased a strong LSTM-based forecasting system and concentrated on predicting daily BTC values. With a lower RMSE of 247.33, the LSTM model with the AR(2) component fared better than the other model. According to this finding, adding autoregressive components to LSTM models might be a practical strategy to improve prediction accuracy. The review advances our understanding of LSTM-based estimating techniques in cryptographic money markets by providing clarity on how these models are generally presented. Stakeholders that want to more precisely and successfully manage market swings will find this information helpful. The researchers looked at three distinct forecasting models for Bitcoin prices: GRU, LSTM, and ARIMA (*Yamak, Yujian & Gadosey, 2019*). With a MAPE of 2.76 percent and an RMSE of 302.53, ARIMA showed the best performance in the studies. The study included two distinct prediction models built using LSTM and Bayesian optimised RNN to estimate the price of BTC (*McNally, Roche & Caton, 2018*). With an RMSE of 8% and an accuracy of 52%, LSTM seems to have performed better. The prediction of bitcoin exchange rates using LSTM, GRU, and Bi-LSTM is assessed in this study. Bi-LSTM performs better than the others and produces the most accurate outcomes. It emphasizes how useful the models are for investors and suggests that future studies look into other aspects that affect cryptocurrency pricing (*Seabe, Moutsinga & Pindza, 2023*). A hybrid LSTM-GRU model is proposed for cryptocurrency price forecasting using time series analysis. Evaluated on Bitcoin, Ethereum, and Ripple datasets, it achieves the lowest MSE and RMSE values, demonstrating high accuracy for future price predictions (*Auliyah, 2024*).

Analysing the cryptocurrency market's past price fluctuations is a crucial stage in the investing process. One of the main tactics that investors have used is building Markov

chains, which comprises generating many decision trees in order to determine which cryptocurrencies have the highest chance of increasing in value when sold (*Lazo et al., 2019*). The writers compared the expected and actual results to determine how effective this was. Since precise forecasting is essential for investment endeavours, this study focusses on three different models that are capable of predicting the future prices of cryptocurrencies. The project aims to construct accurate prediction models by utilizing artificial intelligence approaches and machine learning algorithms. The ultimate objective is to maximize investors' potential for money production in the erratic cryptocurrency markets by providing them with trustworthy tools to make educated judgments.

## MATERIALS AND METHODS

This research aims to forecast the values of BTC, ETH, and LTC using DL methods like LSTM and GRU. The study adheres to a particular process for evaluation purposes, which includes the following steps: (1) gathering historical data for LTC, BTC and ETH; (2) exploring visualization of data; (3) dividing the dataset into testing and training datasets; (4) training different model types; (5) evaluating the models; and (6) contrasting the effectiveness of each model. The section explains the methodologies used during the study's pre-processing and modeling stages. Furthermore, the various results of the prediction plots for some cryptocurrencies have been provided below. Finally, an overview of the complete research, including analysis and implementation, was presented.

### Proposed methodology

In this article, a two-layer network with 100 deep learning layers is built using LSTM and GRU. The writers depended on a credible source known as CryptoDataDownload, which provides detailed historical market data for cryptocurrencies. It delivers detailed organized datasets, which is a vital motivator for undertaking exact analysis in Bitcoin research. The preprocessing step ensures consistent feature scaling, which is essential for efficient model training and improves convergence during the learning process. Various pre-processing methods applied to the cryptocurrency dataset prepared it for deep learning processing. Normalisation eliminates bias and guarantees that a model is properly fitted. MinMaxScaler was applied from sklearn to normalize the dataset, scaling features to a uniform range without distorting value differences. Normalization enhances the models' data handling capabilities, leading to more reliable and accurate outcomes (*Ahsan et al., 2021*; *Gupta, 2021*). For each cryptocurrency, a train:test split ratio of 80:20 was used to ensure feature consistency. Data from August 17, 2017 to February 22, 2023 (80% of the total data) make up the training dataset. On another hand, 20% of the testing dataset's data comes from February 23, 2023 to July 10, 2024. Python 3 was used for the studies, along with pertinent libraries including Matplotlib, Pandas, NumPy, Keras, and scikit-learn. Figure 1 displays the methodology of data preprocessing and selection of models.

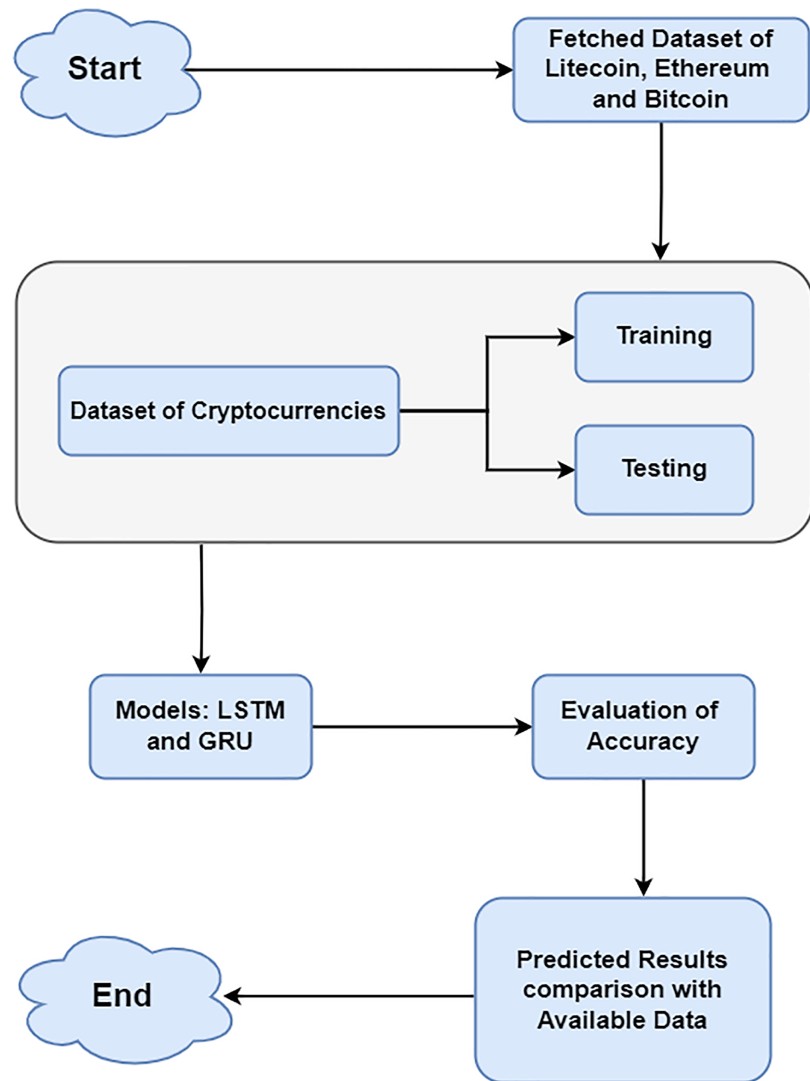

**Figure 1 Methodology of data processing and selection of models.**

## Dataset description

Figures 2–4, which are separated into training and testing datasets, show the daily closing values of three cryptocurrencies, namely ETH, LTC, and BTC. The dataset used in this study is publicly available and was sourced from CryptoDataDownload, specifically from the Binance exchange. To exclude repetitive data from the early days of cryptocurrencies, only historical data from the previous seven years were incorporated. The authors observed that the prices of each currency have generally increased and decreased over the time series.

Bitcoin price prediction graph is shown in Fig. 2, with "Train" (blue) and "Test" (orange) values. Dates from 2021 to 2024 are represented on x-axis while BTC values are displayed on y-axis between 0 and 70,000 USD. A prediction model is trained using historical BTC values over the "Train" period, which runs from early 2021 to late

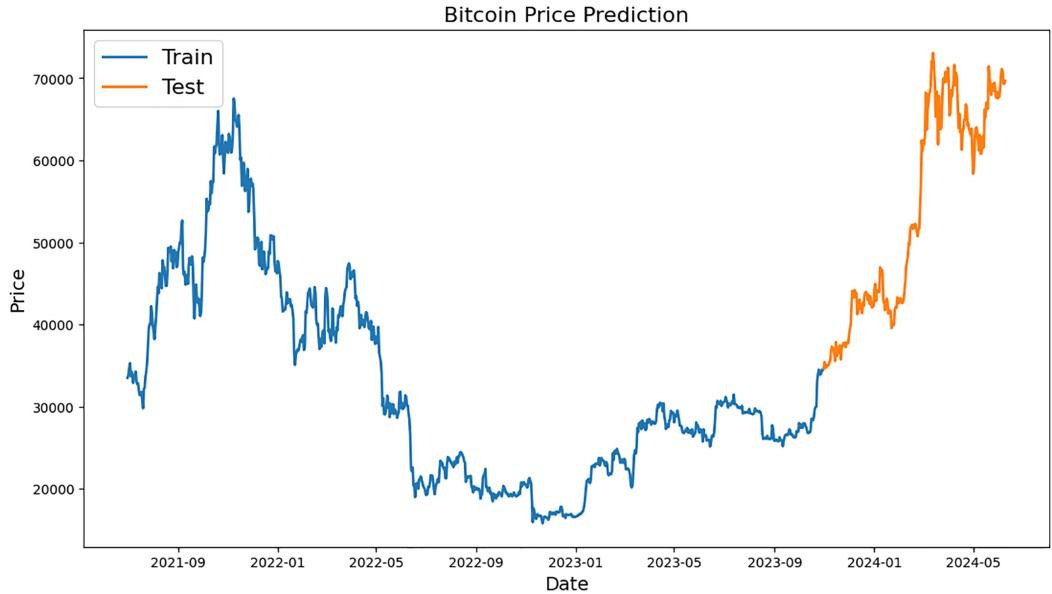

**Figure 2 Example of training and testing for Bitcoin.**

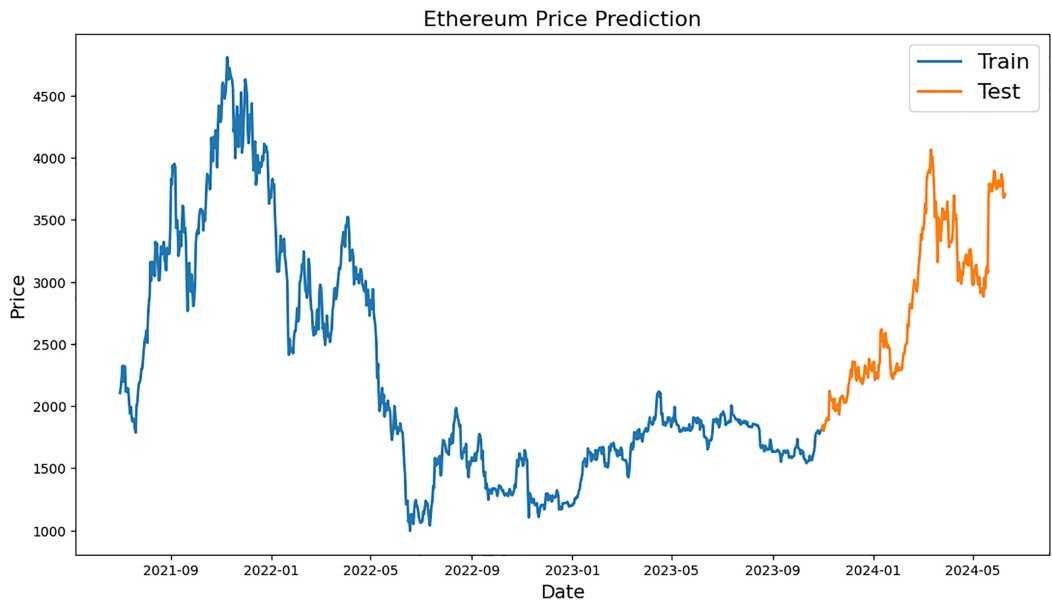

**Figure 3 Example of training and testing for Ethereum.**

2023. When BTC prices peaked in late 2021, they were close to 20,000 USD and by the end of 2023, they had fluctuated and recovered somewhat. Starting in late 2023 and lasting until mid-2024, the "Test" period incorporates pricing projections derived from the model. Prices increased significantly during this time, rising from roughly 20,000 USD to over 70,000 USD, demonstrating significant growth.

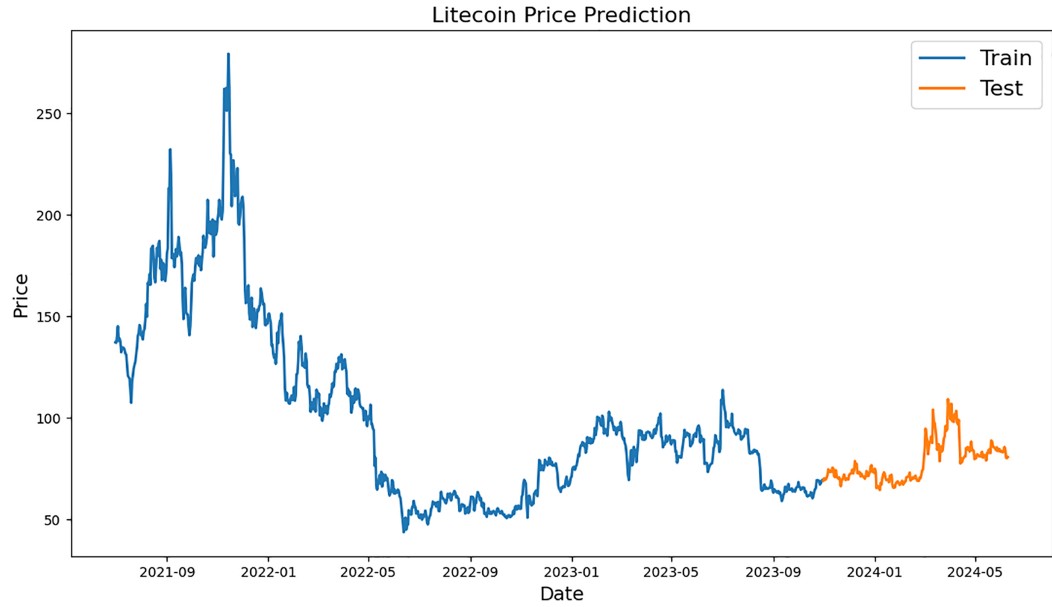

**Figure 4** **Example of training and testing for Litecoin.**

Figure 3 shows Ethereum price forecast graph; orange numbers for "Test" and blue for "Train" The x-axis shows the range of dates from 2021 to mid-2024, and the y-axis shows the prices of Ethereum, which vary from 0 to 5,000 USD. The historical Ethereum price "Train" period chart, which runs from early 2021 to late 2023, is shown below. It is used to train a prediction model. The price of ether fell to roughly $1,000 in the middle of 2022 from a peak of almost $4,500 in late 2021. After that, there were ups and downs until the end of 2023, when there was a brief period of stability. The model price forecasts are accessible from the end of 2023 to the middle of 2024, which is known as the "Test" period. Prices increased significantly at this time, rising from roughly 1,500 USD to over 4,000 USD; this indicated a noticeable upward trend in prices.

A line graph representing the historical and predicted values of LTC during a certain time period is displayed in Fig. 4. The price, which is assumed to be between $50 and $300, is defined by the y-axis, while the x-axis specifies the time period, which runs from mid-2021 to mid-2024. The preparation information ("Train") and the test information ("Test") are the two sections that make up the diagram. The blue training data represent the actual historical values of LTC from mid-2021 to late-2023. Within this time frame, there could be significant swings, with the price reaching a high of over $250 in late 2021 and then generally falling with a lot of volatility. From late 2023 to mid-2024, the orange test data, which represents the anticipated pricing, is displayed. The anticipated prices exhibit little volatility within the specified timeframe, indicating a more stable future price trend for LTC.

Deep learning and statistical analysis rely upon the capability to interpret the distribution of a dataset. Using visual tools, the data's patterns, trends and abnormalities can be quickly identified. The precision of models and decision-making process can both

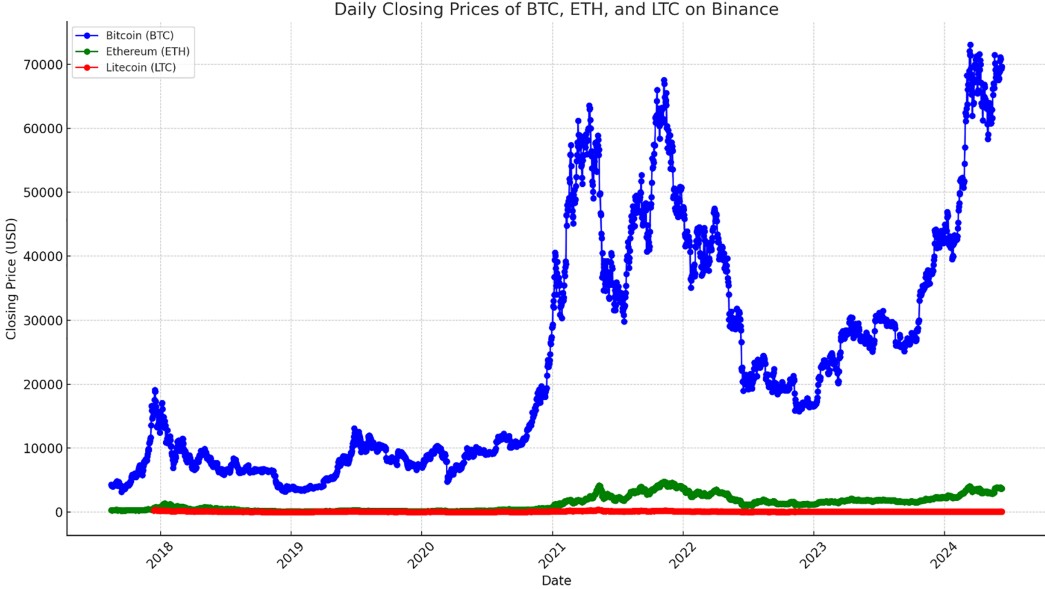

**Figure 5 Time series data with closing prices of LTC, BTC and ETH.**

be improved by these revelations. Additionally, by making complicated information easier to comprehend and available, visualizations may improve stakeholder comprehension and cooperation. The data of time series for LTC, BTC, ETH from August 17, 2017, to June 10, 2024, is displayed in Fig. 5. The time frame was selected to ensure that there were enough dataset items to feed the deep learning models.

A sample of data from the selected cryptocurrency dataset used in this work is shown in Fig. 6. The data properties and distributions of the cryptocurrencies are also shown. A clear picture of the sample data is provided by this visualization, which also serves to draw attention to trends or patterns that could be pertinent to the work.

The correlation matrix shows the correlation coefficient between two cryptocurrencies which tells how changes in single variable's value impact changes in another. When changes in one variable match changes in another, the changes are seen as correlated. So, the variables are considered substantially associated if the value of correlation coefficient is more than 0.5. The closing values of BTC, LTC and ETH show a positive link, according to the correlation matrix in Fig. 7. Because of their strong connection, it is likely that the prices of the other cryptocurrencies will move in the same way if the price of one of them increases or decreases.

### Deep learning algorithms

This section shows two types of DL models, namely, long short-term memory and gated recurrent unit (*Chung et al., 2014*; *Yang, Yu & Zhou, 2020*; *Hochreiter & Schmidhuber, 1997*). LSTMs are more versatile and ideal for complex sequences, whereas GRUs are more computationally efficient.

|   | Unix | Date | Symbol | Open | High | Low | Close | Volume LTC | Volume USDT | tradecount |
|---|------|------|--------|------|------|-----|-------|------------|-------------|------------|
| 0 | 1.717890e+12 | 6/9/2024 | LTCUSDT | 79.96 | 80.71 | 79.49 | 80.49 | 163718.623 | 13125077.48 | 51081 |
| 1 | 1.717800e+12 | 6/8/2024 | LTCUSDT | 80.13 | 80.46 | 78.84 | 79.97 | 257436.338 | 20544533.15 | 66426 |
| 2 | 1.717720e+12 | 6/7/2024 | LTCUSDT | 84.19 | 84.83 | 75.00 | 80.13 | 782793.301 | 63136585.68 | 195737 |
| 3 | 1.717630e+12 | 6/6/2024 | LTCUSDT | 85.44 | 85.84 | 84.16 | 84.19 | 296197.824 | 25200985.06 | 92679 |
| 4 | 1.717550e+12 | 6/5/2024 | LTCUSDT | 83.66 | 85.47 | 83.49 | 85.44 | 302276.545 | 25467926.84 | 76721 |

(a)

|   | Unix | Date | Symbol | Open | High | Low | Close | Volume ETH | Volume USDT | tradecount |
|---|------|------|--------|------|------|-----|-------|------------|-------------|------------|
| 0 | 1.717890e+12 | 6/9/2024 | ETHUSDT | 3681.58 | 3721.52 | 3666.36 | 3706.40 | 103451.1020 | 3.818545e+08 | 287960 |
| 1 | 1.717800e+12 | 6/8/2024 | ETHUSDT | 3678.31 | 3709.50 | 3660.08 | 3681.57 | 140550.8394 | 5.181185e+08 | 364485 |
| 2 | 1.717720e+12 | 6/7/2024 | ETHUSDT | 3813.47 | 3841.39 | 3600.00 | 3678.32 | 362223.3594 | 1.352308e+09 | 894006 |
| 3 | 1.717630e+12 | 6/6/2024 | ETHUSDT | 3866.00 | 3878.60 | 3760.00 | 3813.46 | 237504.1905 | 9.107518e+08 | 525050 |
| 4 | 1.717550e+12 | 6/5/2024 | ETHUSDT | 3810.23 | 3887.47 | 3777.33 | 3865.99 | 273738.6035 | 1.046374e+09 | 645836 |

(b)

|   | Unix | Date | Symbol | Open | High | Low | Close | Volume BTC | Volume USDT | tradecount |
|---|------|------|--------|------|------|-----|-------|------------|-------------|------------|
| 0 | 1.717890e+12 | 6/9/2024 | BTCUSDT | 69310.46 | 69857.14 | 69130.24 | 69648.14 | 9890.56709 | 6.873447e+08 | 575583 |
| 1 | 1.717800e+12 | 6/8/2024 | BTCUSDT | 69355.60 | 69582.20 | 69168.02 | 69310.46 | 9773.82967 | 6.782674e+08 | 714103 |
| 2 | 1.717720e+12 | 6/7/2024 | BTCUSDT | 70799.06 | 71997.02 | 68420.00 | 69355.60 | 35598.45045 | 2.507251e+09 | 1516415 |
| 3 | 1.717630e+12 | 6/6/2024 | BTCUSDT | 71108.00 | 71700.00 | 70117.64 | 70799.06 | 21842.00449 | 1.551468e+09 | 900226 |
| 4 | 1.717550e+12 | 6/5/2024 | BTCUSDT | 70537.83 | 71758.00 | 70383.66 | 71108.00 | 28703.18082 | 2.040074e+09 | 1055115 |

(c)

**Figure 6  A snippet exhibiting data sample of (A) BTC dataset; (B) ETH dataset; (C) LTC dataset.**

## Long short-term memory

An improved RNN architecture, namely, LSTM is intended primarily to tackle the vanishing gradient problem and deal with the problem of long term dependence. An information-regulatory system built into LSTMs allows information to be stored for long periods (*Yang, Yu & Zhou, 2020*). To put it briefly, the architecture of LSTM consists of many memory blocks that are connected sub-networks recurrently. Every single one of these memory blocks is essential to the network's overall operation. Firstly, they are in charge of preserving the network's state across time. This indicates that the memory blocks have a long-term retention capacity, which is critical for activities requiring comprehension of context across lengthy periods. Secondly, the information that moves between the cells is controlled by the memory blocks. They manage which data is retained, updated or removed at each time step, so that the network can efficiently recognize and learn from significant patterns without being influenced by insignificant information. These dual functions provide LSTMs the ability to manage long-term dependencies and alleviate problems like as the gradient problem, which makes them very efficient for a variety of sequential data applications.

**Peer**J Computer Science

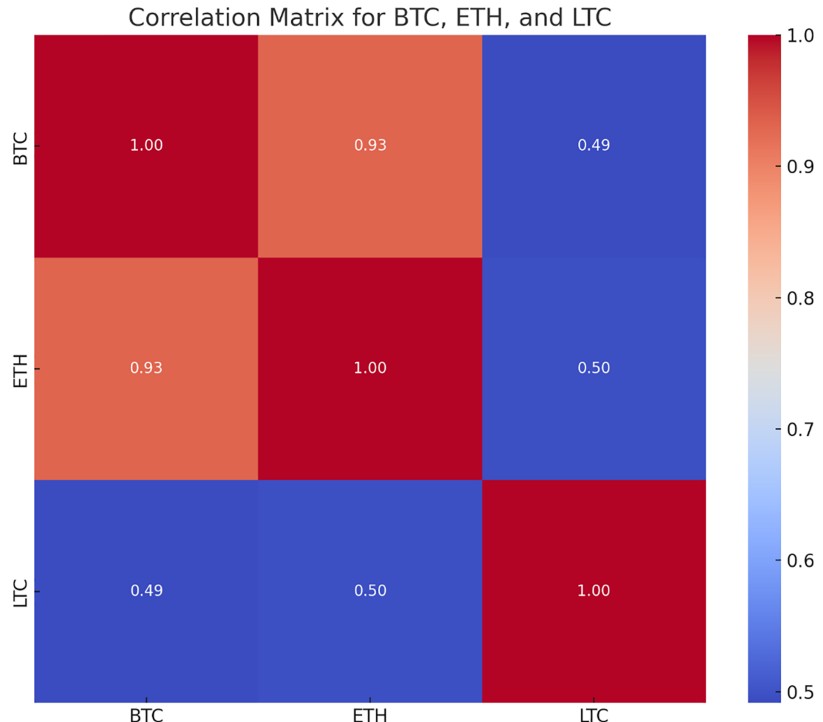

**Figure 7 Correlation matrix for LTC, BTC and ETH.**

An LSTM model is built using a configuration of 100 neurons for network to recognize complex patterns in input. A batch size of 32 was used during 20 epochs of training in order to balance computational effort and learning efficiency. A dropout rate of 0.2 is used during training, arbitrarily deactivating 20% of neurons to reduce overfitting. Adam is optimizer of choice for the optimization process and has an adaptive learning rate that makes it highly effective. Since the LSTM's design includes a linear activation function, which further guarantees extremely accurate continuous value model prediction, it would be ideal for forecasting time series and other continuous data.

The chain structure of the LSTM architecture comprises four neural networks and various memory blocks known as cells. The cells store information, and the gates carry out memory processing. Three gates are present:

- Forget gate: The forget gate eliminates data that is no longer relevant in the cell state. The gate receives two inputs, $x_t$ (input at that specific moment) and $h_{t-1}$ (output from the preceding cell), which are multiplied by weight matrices before bias is added. An activation function sigmoid ($\sigma$) is applied to the outcome, producing a binary output. A piece of information is lost if the output for a certain cell state is 0, but it is kept for later use if the output is 1. The forget gate's Eq. (1) is:

$$f_t = \sigma\left(W_f.[h_{t-1}, x_t] + b_f\right) \tag{1}$$

where

$W_f$ is weight matrix connected to the forget gate

$b_f$ is forget gate's bias.

- Input gate: The input gate is responsible for adding valuable information to the cell state. Initially, the sigmoid function is used to regulate the data, and the inputs $h_{t-1}$ and $x_t$ are used to filter the values to be remembered in a manner akin to the forget gate. The tanh function is then used to construct a vector that contains all of the possible values from $h_{t-1}$ and $x_t$ and has an output ranging from −1 to +1. Finally, the useful information is obtained by multiplying the vector values by the regulated values. The input gate's Eq. (2) is:

$$i_t = \sigma(W_i.[h_{t-1}, x_t] + b_i). \tag{2}$$

- Output gate: The output gate is responsible for obtaining valuable information from the current cell state to be displayed as output. The tanh function is first used to the cell to create a vector. The data is then controlled by the sigmoid function and filtered by the values that need to be retained using inputs $h_{t-1}$ and $x_t$. Finally, to be delivered as an output and input to the following cell, the vector values and the controlled values are multiplied. The output gate's Eq. (3) is:

$$o_t = \sigma(W_o.[h_{t-1}, x_t] + b_o). \tag{3}$$

## Gated recurrent unit

Based on the architecture with LSTM networks, GRU, an extension of RNN was introduced (*Wang, Jiang & Luo, 2016*). Like LSTMs, GRUs are also built to handle arbitrary input sequences of the same length and continue a state that contains information about past inputs. This means that GRUs has also different operations inside when compared to LSTMs. LSTMs need more gates and an internal memory cell, whereas GRUs have a simpler way of controlling the information flow. They have both a reset gate and one update gate that together decide what should be forgotten GRUs are simpler than LSTMs as they may be easier to train. Despite their simple architecture, GRUs can perform as well as LSTMs in many application areas (*Yang, Yu & Zhou, 2020*).

So, the Penn Treebank dataset showed that GRUs have outperformed LSTM networks for language modeling. This result goes to prove the superiority of GRUs for noisier language models. Also, a comparison of natural language processing (NLP) models across different benchmarks viewed GRUs as competitive to both LSTMs and CNNs (*Wang, Jiang & Luo, 2016*). Highlighting the adaptability and strong performance of GRUs across a range of NLP tasks. Catching long-range connections in consecutive information more effectively than normal RNNs is one of GRUs' principal benefits. In view of the ongoing info and the organization's express, the update and reset entryways in GRUs' plan empower them to recall or dispose of verifiable information specifically. Selective retention is essential for activities that require ability to retain and apply knowledge from long sequences. GRUs, for instance, excel in language translation because they are able to effectively manage long-term dependencies, which are essential when preserving context

over lengthy text sequences. In light of this element, GRUs are particularly appropriate for a large number of successive information handling applications, and in certain circumstances, they even outflank clearer RNNs and LSTMs (*Yang, Yu & Zhou, 2020*).

With 100 neurons in the design, the network can capture complicated sequences and temporal patterns in the data to build a robust GRU model. With 32 batch size, the training procedure will span 20 epochs in order to balance resource usage and training effectiveness. In order to avoid overfitting, the Dropout of 0.2 suggests that 20% of neurones would be switched off during training. Because of the Adam optimizer's adaptive learning rate and effectiveness when dealing with sparse gradients, it is used for optimisation. Because the GRU design employs an additional linear activation in addition to other methods, it is especially well-suited for tasks like time series forecasting or any other task that calls for a model to predict continuous values with accuracy. The reset gate and the update gate are the two gating mechanisms that the GRU has. While the update gate decides how much of the new input should be used to update the hidden state, the reset gate decides how much of the prior hidden state should be forgotten. The updated hidden state is used to calculate the GRU's output. The following Eqs. (4)–(7) are used to determine a GRU's reset gate ($r_t$), update gate ($z_t$), hidden state ($h_t$) and candidate hidden state ($h_{tt}$):

$$r_t = \sigma(W_r.[h_{t-1}, x_t]) \tag{4}$$
$$z_t = \sigma(W_z.[h_{t-1}, x_t]) \tag{5}$$
$$h_t = (1 - z_t) * [h_{t-1}] + z_t * h_{tt}) \tag{6}$$

where

$$h_{tt} = tanh(W_h.[r_t * h_{t-1}, x_t]) \tag{7}$$

$W_r$, $W_z$, $W_h$ are weight matrices.

## Experimental setup and hyperparameters tuning for models

The algorithms were developed in Python 3 and ran using Google Colab on an HP system. The Google Colab virtual machine uses a T4 GPU, Python 3, 16 GB GPU memory, 12.7 GB RAM, and 2 vCPUs, running Ubuntu 20.04 LTS with ~100 GB storage. In order to achieve the best possible performance for the models, this section presents the hyperparameters that were used in this investigation. Finding the ideal model architecture parameters that could be used to anticipate prices is known as hyperparameter tuning. The process of tuning involves choosing an optimal collection of hyperparameters while taking the learning algorithm into account (*Sharma & Babbar, 2023*; *Juneja et al., 2024*). Hyperparameter tuning was performed using a systematic approach to ensure valid comparisons between models. It was performed using random search evaluating combinations through RMSE and MAE. The research parameters are enumerated as follows:

- GRU neurons: 100 neurons, optimizing the model's capacity for learning temporal dependencies.

- Epochs: 20, with early stopping applied to prevent overfitting.
- Batch size: 32, balancing training efficiency and model stability.
- Loss function: MSE, RMSE, MAE and MAPE chosen for its effectiveness in regression tasks.
- Dropout rate: 0.2, used to mitigate overfitting by introducing regularization.
- Optimizer: Adam, selected for its adaptive learning rate and robust optimization performance.
- Activation function: Linear, suitable for the continuous nature of cryptocurrency price predictions.

## Performance metrics

The performance of data is statistically measured by a number of metrics related to it. The metrics for deep learning models that will be the focus for this study are MSE, RMSE, MAE and MAPE (*Garg et al., 2022*; *Kazeminia, Sajedi & Arjmand, 2023*). The goal is to evaluate various models and select the best based on the performance metrics that follow:

- Mean squared error: It estimates the average squared difference among real and expected values. It offers a quadratic loss metric that penalizes greater mistakes more harshly. The performance of the prediction model improves with a reduced MSE value. MSE helps improve the accuracy of the GRU and LSTM models for Bitcoin, Litecoin, and Ethereum by efficiently penalizing significant differences in forecast prices. The mathematical representation for MSE is mentioned in Eq. (8):

$$MSE = \frac{1}{n} \sum_{t=1}^{n} (A_t - \hat{A}_t)^2 \tag{8}$$

- Root mean squared error: The measurement of the discrepancies between values that a model predicts and values that are actually observed is called the root mean square error. It is simpler to read since it gives an error measure in the same units as the target variable. The prediction model performs better with lower RMSE value. By explicitly representing the error magnitude in the same units as the values of Bitcoin, Litecoin, and Ethereum, RMSE offers an interpretable indicator of forecast accuracy for cryptocurrency prices. The mathematical representation for RMSE is mentioned in Eq. (9):

$$RMSE = \sqrt{\frac{1}{n} \sum_{t=1}^{n} (A_t - \hat{A}_t)^2} \tag{9}$$

- Mean absolute error: It estimates the average absolute difference among real and expected values. It offers a linear loss measure that does not severely penalize big mistakes. The prediction model performs better when the MAE value is lower. Without unduly penalizing significant errors, MAE provides a strong evaluation of average

prediction error, which aids in assessing how consistently GRU and LSTM models predict Bitcoin prices. Equation (10) describes the MAE mathematical representation:

$$MAE = \frac{1}{n} \sum_{t=1}^{n} |A_t - \hat{A}_t|. \tag{10}$$

- Mean absolute percentage error: It estimates the average absolute percentage difference among real and expected values. With a lower MAPE value, the prediction model performs better. Even if the price ranges of Bitcoin, Litecoin, and Ethereum vary, MAPE makes it easier to compare their price predictions, allowing for a scale-independent assessment of model performance. Equation (11) identifies the mathematical representation of MAPE:

$$MAPE = \frac{100}{n} \sum_{t=1}^{n} \left| \frac{A_t - \hat{A}_t}{A_t} \right|. \tag{11}$$

## RESULTS

Keras, Sklearn and TensorFlow are various Python libraries used in the proposed DL model. The tools and frameworks expected for making and refining the models were made accessible by these libraries. Python 3.9 was utilized to carry out the calculations, ensuring that it would work with the latest enhancements to the Python programming language. An interactive platform for coding, troubleshooting, and result visualization was provided by the programming environment known as Colab. Predicting the actual values of such cryptocurrencies, including BTC, ETH, and LTC, was the challenge. Numerous deep learning models have been developed and similarly examined for that reason. The table provides error values for each model, and comparing the outcomes.

LTC prediction metrics utilizing LSTM and GRU models are displayed in Table 1. With a lower MSE (0.00029) and RMSE (0.01705), GRU performs better than LSTM, suggesting higher prediction accuracy. GRU's MAPE (0.08037) is marginally higher than LSTM's (0.07653), and its MAE (0.01272) is marginally higher than LSTM's (0.01201). All things considered, GRU offers a more trustworthy prediction model with notable gains in important error metrics, especially MSE and RMSE. The ETH prediction performance for LSTM and GRU is shown in Table 2. With a lower MSE (0.00093) and RMSE (0.03051) than LSTM, which has equivalent values of 0.00124 and 0.03532, respectively, GRU achieves improved accuracy. In addition, GRU shows somewhat lower MAPE (0.04415) and MAE (0.02131) than LSTM. Overall, GRU performs better than LSTM, demonstrating its capacity to more accurately model ETH data with lower prediction error rates on all measures. The BTC prediction outcomes are evaluated in Table 3. With a substantially reduced MSE (0.00106 *vs.* 0.00455) and RMSE (0.03258 *vs.* 0.06749), GRU performs noticeably better than LSTM. Moreover, GRU shows better prediction accuracy with a significantly lower MAPE (0.03540) and MAE (0.02342). The outcomes demonstrate

**Table 1 Performance results for LTC.**

| Currency | MSE | RMSE | MAPE | MAE |
|---|---|---|---|---|
| LSTM | 0.00032 | 0.01799 | 0.07653 | 0.01201 |
| GRU | 0.00029 | 0.01705 | 0.08037 | 0.01272 |

**Table 2 Performance results for ETH.**

| Currency | MSE | RMSE | MAPE | MAE |
|---|---|---|---|---|
| LSTM | 0.00124 | 0.03532 | 0.05075 | 0.02471 |
| GRU | 0.00093 | 0.03051 | 0.04415 | 0.02131 |

**Table 3 Performance results for BTC.**

| Currency | MSE | RMSE | MAPE | MAE |
|---|---|---|---|---|
| LSTM | 0.00455 | 0.06749 | 0.09162 | 0.05627 |
| GRU | 0.00106 | 0.03258 | 0.03540 | 0.02342 |

GRU's effectiveness in identifying trends and reducing prediction metric errors, making it the superior model for predicting the price of Bitcoin.

The MSE, MAPE, RMSE and MAE scores were utilized to assess the performance of LSTM model over LTC, ETH, and BTC data are displayed in Table 1. Among them, the lowest RMSE was 0.01799 for LTC; the MAE was 0.01201 for LTC; the MSE was 0.00032 for LTC; and the MAPE was 0.05075 for ETC for the model that produced the most accurate forecast.

Table 2 shows the findings for the GRU model on BTC, LTC, and ETH data, which were analyzed using MSE, MAPE, RMSE, and MAE. It has the lowest LTC RMSE of 0.01705, the lowest MAE of 0.01272, the highest LTC MSE of 0.00029, and the greatest BTC MAPE of 0.03540.

As a result, GRU had the lowest error numbers and hence had the best performance. Given its outstanding accuracy in predicting Bitcoin value, the model will provide a useful tool for financial forecasting. The thorough analysis and comparison proves GRU as the best model in this study, yielding accurate predictions based on the available data.

## Results for LTC

Figures 8A and 8B displays a graph which contrasts the actual and anticipated values of LTC for GRU and LSTM respectively. Time is represented by x-axis, while pricing, which ranges from about $70 to $110, is represented by y-axis. Two lines are plotted, namely, the expected price line is shown in red, and the true price line is shown in blue. When the two lines on the graph first nearly match, it indicates that the early prognosis was true. The true price shows notable fluctuations over time, with highs and lows, especially in the center of

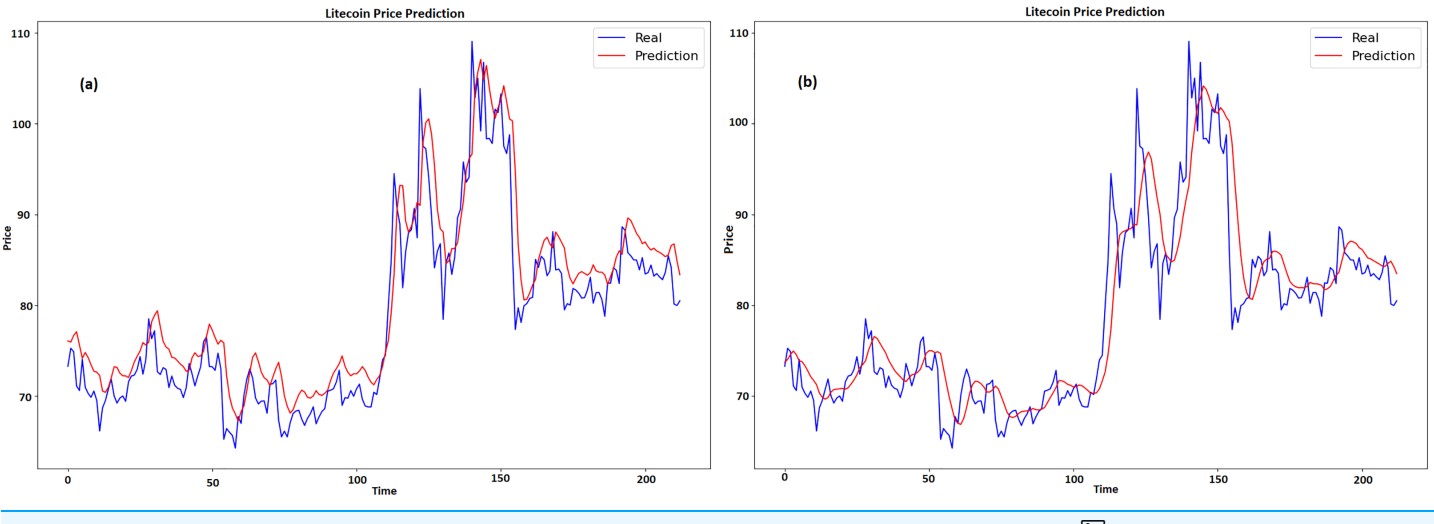

**Figure 8 Real and predicted price of LTC (A) GRU (B) LSTM.**

the graph when it crosses $100. The anticipated price has a comparable pattern with more gradual swings, suggesting that the predictive model tends to average out large variances in the actual pricing.

Figures 9A and 9B illustrates a graph which shows the loss values over the course of 20 epochs throughout training and testing stages of model for GRU and LSTM respectively. The y-axis displays loss that ranges from 0 to 0.0175 and 0 to 0.014 respectively, while the x-axis depicts the epochs, which range from 0 to 20. There are two lines plotted, namely, an orange line for test loss and a blue line for training loss. After the first few epochs, the training loss steadies at roughly 0.0025 after beginning unusually high at 0.0175. In contrast, the test loss is substantially smaller at first and stays low throughout the epochs, averaging approximately 0.0003. The model learns from training data fast, as proven by large decrease in training loss in early epochs. The capability of model to effectively generalize to new data is indicated by its continuously low test loss. It also maintains low error rates during testing, indicating efficient training without overfitting.

## Results of ETH

Figures 10A and 10B compares the actual and expected values of Ethereum over time for GRU and LSTM, respectively. Time is denoted by x-axis, whereas price is shown on y-axis. The projected prices are displayed in red and the actual prices are displayed in blue. At first, the forecast closely tracks the actual price, catching both the general trend and the swings. There are very little variations, particularly during abrupt fluctuations in price, since the model seeks to level out the highs and lows. This pattern can be seen at the 100 and 150 time points, where the forecasts do not completely reflect big spikes and falls in the real prices. When predicting the overall direction and size of price fluctuations, the model does a good job. The model's resilience is seen in the later portion of the graph, as it adapts to sudden price rises and ensuing stability. The model's ability to anticipate the overall

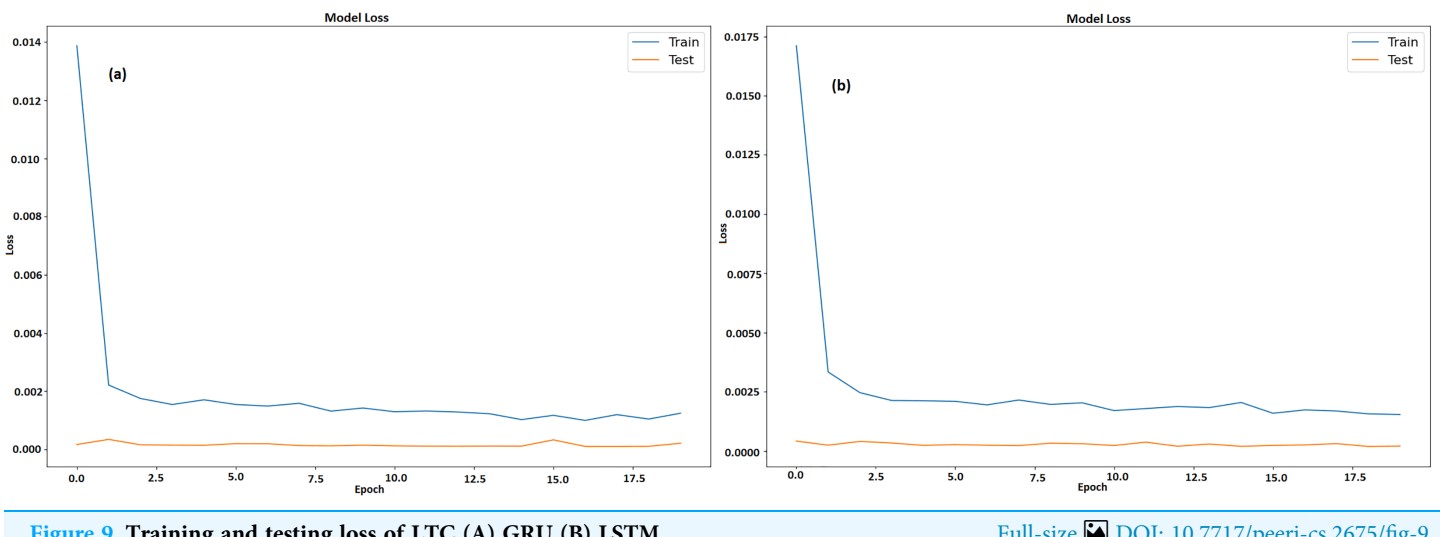

**Figure 9** **Training and testing loss of LTC (A) GRU (B) LSTM.**

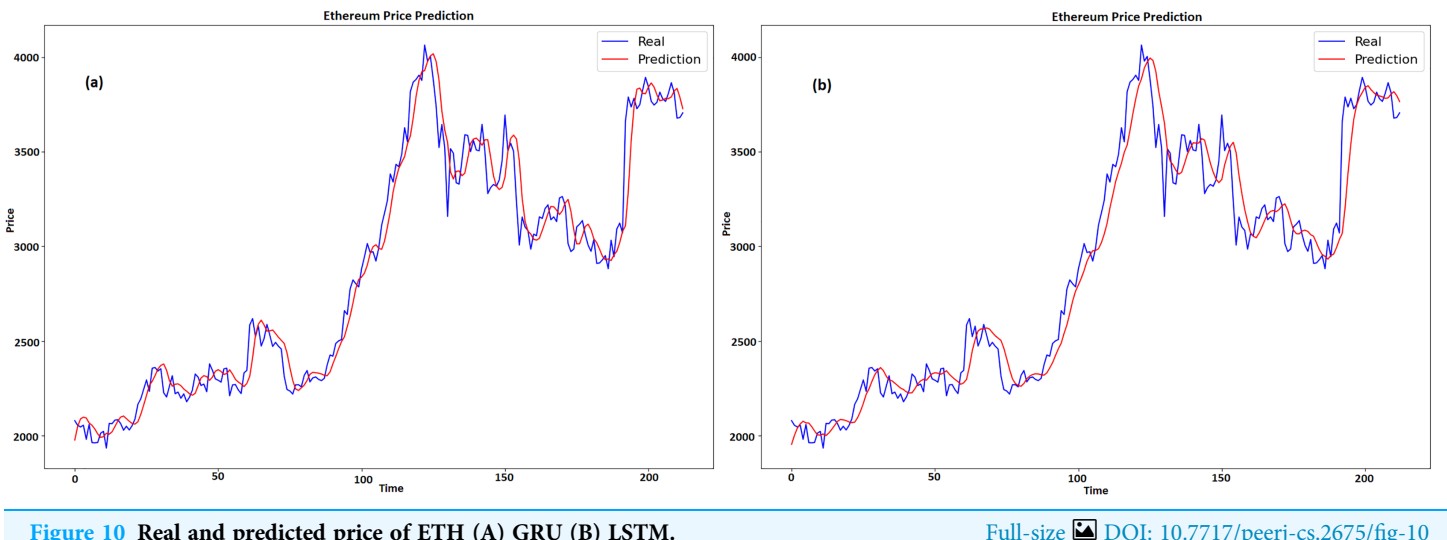

**Figure 10** **Real and predicted price of ETH (A) GRU (B) LSTM.**

trend, even with small errors during turbulent times, suggests that it may have some use in projecting Ethereum values.

Figures 11A and 11B displays a model's training and testing loss during a period of 20 epochs for GRU and LSTM respectively. The epoch count is shown on x-axis whereas the loss is denoted by y-axis. The testing loss is displayed in orange, while the training loss is displayed in blue. The training loss is considerable at the start (epoch 0), suggesting that the model has difficulty fitting the data at first. But in the first few epochs, the training loss rapidly drops and stabilizes at a lower value around epoch 5. Following this, there are a few slight variations in the training loss but overall it stays low, meaning the model is successfully learning from the training set. On the other hand, with very modest

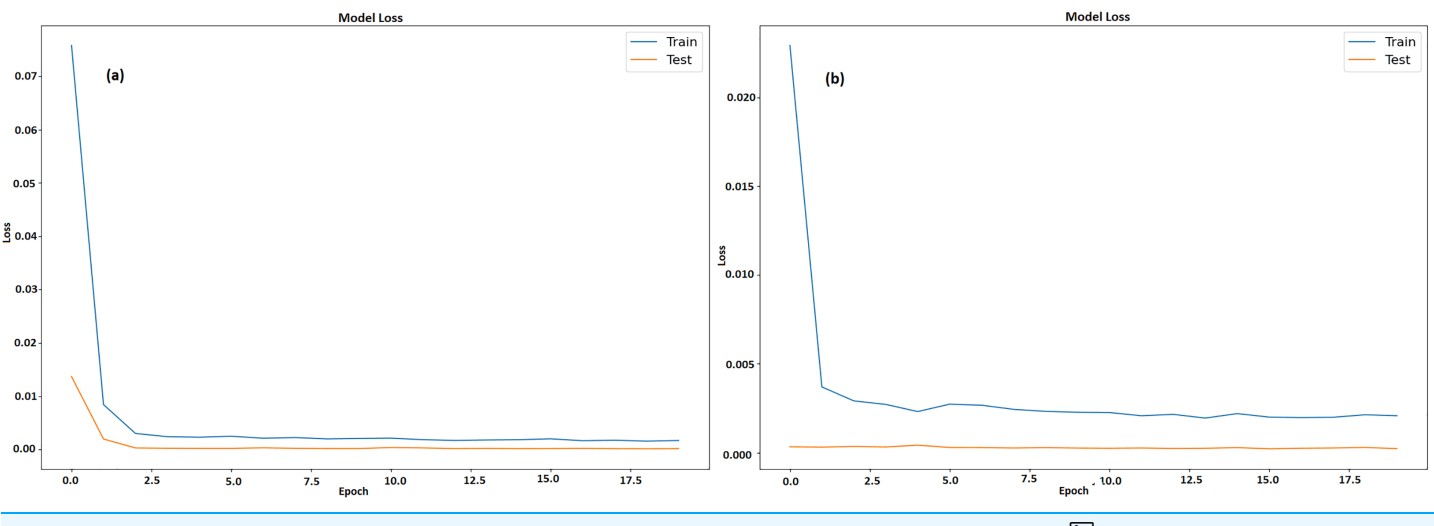

**Figure 11  Training and testing loss of ETH (A) GRU (B) LSTM.**     

fluctuations, the testing loss starts low and stays low throughout the epochs. This consistency indicates that it is not overfitting on training set and can generalize fit to fresh data. The model performs well and continues to generalize with low loss values on both training as well as testing data sets.

## Results of BTC

Figures 12A and 12B shows the historical real *vs* expected values of Bitcoin for GRU and LSTM respectively, with time represented by x-axis and price by y-axis. The projected prices are displayed in red and the actual prices are displayed in blue. At first, the forecast captures the overall increasing trend and matches the real price. But with time, especially during times of significant price fluctuations, the projected costs start to fall short of the actual prices. The real prices exhibit enough volatility that the projections are unable to completely reflect at the 100 and 150 time points. A portion of the real prices' higher peaks and lower troughs are missed by the prediction line, which has a tendency to smooth out price variations. This suggests that although the model does a good job of capturing the general trend and direction, it has trouble correctly predicting the size of abrupt price swings. Though it might use some tweaks to better manage volatility, the model does a respectable job of protecting the overall direction of Bitcoin values despite these disparities.

Figures 13A and 13B displays a machine learning model's testing and training loss over a period of 20 epochs for GRU and LSTM respectively. The epoch count is shown by x-axis whereas the loss is denoted by y-axis. The model learns fast from training data, as seen by the somewhat large initial training loss that immediately lowers during the first few epochs. Following this sharp decline, the training loss steadily levels out and stays low, indicating that model is becoming better and adjusting to the training set more effectively. Results acquired from this study demonstrate that GRU model outperformed the LSTM model when it came to predicting the prices of all cryptocurrencies.

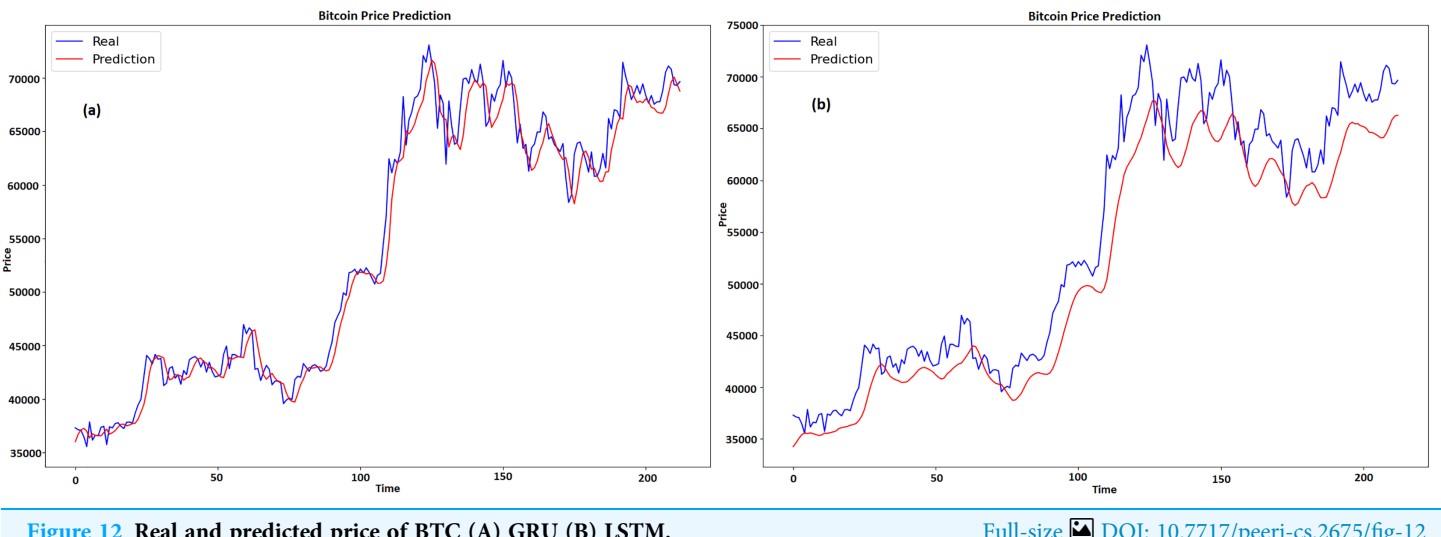

**Figure 12  Real and predicted price of BTC (A) GRU (B) LSTM.**     

**Figure 13  Training and testing loss of BTC (A) GRU (B) LSTM.**     

## DISCUSSION

For cryptocurrency prediction, this study's proposed model can be regarded as acceptable and reliable. Table 4 shows how this article's proposed model comparison to other models of the literature.

GRU and LSTM were used for cryptocurrency price prediction because they can handle complicated sequential data while overcoming standard RNN constraints such as vanishing gradients. LSTM's gated methods enable excellent long-term dependency modeling, but GRU provides equivalent performance with less computing complexity. Unlike transformers, which are resource-intensive, GRU and LSTM are suitable for small datasets and latency-sensitive jobs. These models achieve a mix between strong temporal

**Table 4 Comparison of proposed model and previous work.**

| Authors | Cryptocurrency | Methods | MAPE | RMSE |
|---|---|---|---|---|
| *Hansun, Wicaksana & Khaliq (2022)* | BTC-USD | LSTM | 0.042 | 2,518.02 |
| | | GRU | 0.038 | 1,777.31 |
| | ETH-USD | LSTM | 0.064 | 150.09 |
| | | GRU | 0.059 | 151.62 |
| *Ozturk Birim (2022)* | BTC-USD | LSTM | 0.040 | 2,350.53 |
| | | GRU | 0.053 | 3,223.01 |
| | ETH-USD | LSTM | 0.047 | 183.84 |
| | | GRU | 0.047 | 181.03 |
| *Wu et al. (2018)* | BTC-USD | LSTM | 2.701 | 256.41 |
| | | Hybrid | 2.533 | 247.33 |
| Our approach | BTC-USD | LSTM | 0.0916 | 0.0674 |
| | | GRU | 0.0354 | 0.0325 |
| | ETH-USD | LSTM | 0.0507 | 0.0353 |
| | | GRU | 0.0441 | 0.0305 |
| | LTC-USD | LSTM | 0.0765 | 0.0179 |
| | | GRU | 0.0803 | 0.0170 |

modeling and computing efficiency, making them excellent for forecasting Bitcoin, Litecoin, and Ethereum values. The outcomes show that the GRU model outperforms the LSTM model in terms of forecasting the prices of BTC, ETH, and LTC. GRU consistently produces lower error levels across all cryptocurrencies, according to an examination of the two models using error metrics like RMSE, MAE, MSE, and MAPE. For example, compared to LSTM, the GRU model obtains the lowest RMSE of 0.01705 for LTC and a lower MAE of 0.01272 for LTC, compared to 0.01201 for LSTM. These findings demonstrate that GRU performs better in identifying price patterns and providing more precise forecasts. Apart from the reduced error metrics, the loss values displayed throughout the training and testing stages clearly demonstrate the GRU model's superior generalization capabilities. While the test loss is consistently low throughout, demonstrating successful learning and avoiding overfitting, the training loss of GRU rapidly decreases after a few epochs and stabilizes at a low value. LSTM, on the other hand, shows a larger test loss, indicating less effective generalization to unknown data.

Although it has trouble predicting the entire magnitude of abrupt price spikes or decreases, the GRU model also does a good job at capturing the general trend of price swings for BTC, ETH, and LTC. GRU tends to smooth out excessive changes, which is especially evident when comparing the anticipated and actual values in figures for all cryptocurrencies. The study emphasizes how much better GRU is at predicting Bitcoin prices in terms of both accuracy and generalization. According to these results, GRU may be a strong model for financial forecasting that provides more accurate and consistent outcomes in erratic markets such as cryptocurrency.

## LIMITATION

- Restricted cryptocurrency scope: Because the study primarily examines the three most important cryptocurrencies—BTC, ETH, and LTC—it may be unable to reliably forecast how other, less liquid, or new cryptocurrencies will behave.

- Absence of feature diversity: If the model just considers historical prices or currency rates, it may neglect important variables such as transaction volume, market sentiment, or macroeconomic data.

- Lack of exogenous factors: The model may not account for external factors that have a significant impact on Bitcoin pricing, such as regulatory changes, social media influence, or worldwide financial events.

- Limited comparisons to other models: This study does not analyze traditional statistical techniques or simpler, non-neural network models such as ARIMA, which could offer light on the relative benefits of LSTM and GRU.

- Metrics for Evaluation: The study focusses primarily on the MAE, MSE, MAPE and RMSE metrics, which may not truly reflect how well the model performs in real-world trading settings where risk and profit are more relevant factors.

## CONCLUSIONS AND FUTURE SCOPE

Two architectures of recurrent neural networks, namely, LSTM and GRU were utilized in this work that predict the BTC, LTC, and ETH values, three distinct cryptocurrency types. The four metrics used to evaluate the models' performance were MSE, RMSE, MAPE and MAE. The setup includes data normalization using MinMaxScaler and model evaluation using a variety of metrics and both models were trained using Python libraries like Keras and TensorFlow. GRU model performed better than LSTM for ETH and BTC, as it has lower error metrics in most categories (MAPE values-0.03540 for BTC, 0.04415 for ETH, and 0.08037 for LTC). For ETH, LTC, and BTC, the RMSE for GRU model was 0.03051, 0.01705, and 0.0325, respectively. In GRU model, the MSE for BTC was 0.00106, ETH was 0.00093, and LTC was 0.00029. For BTC, ETH, and LTC, the GRU model's MAE was 0.02342, 0.02131 and 0.01272, respectively. Therefore, based on the provided metrics, the GRU model is generally the best choice. The lower and more consistent test loss relative to training loss shows that the GRU model can handle unseen data more efficiently, and it also showed a faster convergence rate and improved generalization to new data. The experimental outcomes proved that artificial intelligence is reliable and satisfactory for prediction of cryptocurrency. In future, utilizing pre-trained models on similar financial datasets may improve the predictive capabilities of the models when transfer learning techniques are used. The impact of market sentiment analysis, macroeconomic indicators and social media trends on cryptocurrency prices could be investigated to improve prediction accuracy in further research.

### Funding

This work was supported by the Basic Science Research Program under Grant 2020R1I1A3069700, and by the Technology Development Program of MSS under Grant S3033853. The funders had no role in study design, data collection and analysis, decision to publish, or preparation of the manuscript.

### Grant Disclosures

The following grant information was disclosed by the authors:
Basic Science Research Program: 2020R1I1A3069700.
Technology Development Program of MSS: S3033853.

### Competing Interests

The authors declare that they have no competing interests.

### Author Contributions

- Ramneet Kaur conceived and designed the experiments, performed the experiments, analyzed the data, performed the computation work, prepared figures and/or tables, authored or reviewed drafts of the article, and approved the final draft.
- Mudita Uppal conceived and designed the experiments, performed the experiments, performed the computation work, prepared figures and/or tables, authored or reviewed drafts of the article, and approved the final draft.
- Deepali Gupta conceived and designed the experiments, performed the experiments, performed the computation work, authored or reviewed drafts of the article, and approved the final draft.
- Sapna Juneja conceived and designed the experiments, performed the experiments, performed the computation work, authored or reviewed drafts of the article, and approved the final draft.
- Syed Yasser Arafat conceived and designed the experiments, analyzed the data, authored or reviewed drafts of the article, and approved the final draft.
- Junaid Rashid conceived and designed the experiments, analyzed the data, authored or reviewed drafts of the article, and approved the final draft.
- Jungeun Kim conceived and designed the experiments, analyzed the data, authored or reviewed drafts of the article, and approved the final draft.
- Roobaea Alroobaea conceived and designed the experiments, performed the experiments, performed the computation work, authored or reviewed drafts of the article, and approved the final draft.

### Data Availability

The data used in this study are available at CryptoDataDownload: https://www.cryptodatadownload.com/data/binance. The dataset includes historical cryptocurrency

market data from the Binance exchange, including prices, volume, and other relevant trading metrics.

The code is available in the Supplemental Files.

## Supplemental Information

Supplemental information for this article can be found online at http://dx.doi.org/10.7717/peerj-cs.2675#supplemental-information.

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
