# Peer review of "Development of a cryptocurrency price prediction model: leveraging GRU and LSTM for Bitcoin, Litecoin and Ethereum"

_PeerJ Computer Science, doi:10.7717/peerj-cs.2675_

## Round 0.1 · original submission · Major Revisions

Dear Authors,

Thank you for submitting your article. Feedback from the reviewers is now available. It is not recommended that your article be published in its current format. However, we strongly recommend that you address the issues raised by the reviewers, especially those related to readability, basic reporting, experimental design and validity, and resubmit your paper after making the necessary changes. By the way, some paragraphs are too long to read. They should be divided into two or more for readbility and comprehensibility.

Warm regards,

Reviewer 1 ·

Basic reporting

Dear Authors,

After carefully reviewing your manuscript, I can confidently state that your proposed work demonstrates novelty and has some potential. However, there are several concerns that should be addressed prior to publication.

Avoid using terms such as novel and new. It can be assumed that the work meets a certain level of novelty, given it is under consideration for publication.

Refrain from using camel case for terms where possible. Introducing abbreviations is helpful and has been done correctly in the abstract. However, if certain terms are used only once, their abbreviations can be omitted.

The keywords should be narrowed down to more specific terms. Additionally, they should include terms not already present in the title of the manuscript.

There appears to be an issue with the indentation in lines 53 and 54 that should be resolved.

The introduction should be more concise. Breaking it into shorter paragraphs (5-10 sentences each) would improve readability.

The manuscript clearly outlines the contributions of the work and its structure, which is commendable.

Abbreviation usage needs attention. For example, terms like "DL" and "ML" are introduced in line 148 but are used earlier (line 101). Abbreviations must be introduced at their first occurrence and used consistently throughout the manuscript.

The related works section provides a comprehensive overview of the literature. However, The structure of the paragraphs could be improved by breaking them into smaller, logical sections. This would enhance clarity and flow.

The descriptions of LSTM and GRU architectures should be expanded significantly. Provide a detailed mathematical explanation of their gated structures. Highlight what makes these architectures unique.

Justify the choice of algorithms over traditional techniques like RNNs, moving averages, or emerging approaches such as transformers.

The description of the simulation environment should be reworked. The configuration of the machine used to run experiments on Google Colab is presented, but is irrelevant to the simulations, as they are carried out on the cloud. Instead the authors should focus their description on the the Google Colab virtual machine (ei type of TPU/GPU, ram limits etc).

Some terms in the performance metrics are not adequately explained.

The presentation of results should be improved and aligned with standard practices in the literature.

While descriptive results are provided, comparative tables in the appropriate table environment should be included for better clarity and readability.

Experimental design

Hyperparameter tuning is mentioned, but the techniques used for selecting parameters are unclear and need to be explained in detail. This is crucial for ensuring a valid comparison between models.

While parameters are listed, their respective constraints are not. Please clarify these constraints.

Parameter selections for the final models are also not provided. As such the simulations and claimed results cannot be independently verified further.

Validity of the findings

While the general scores seem somewhat viable, due to a lack of parameter configuration values in the manuscript, findings cannot be validated and the results cannot be replicated independently.

The authors need to include additional information of the simulation configuration in order to provide further validity to the presented claims. Statistical validations would also benefit the validity of the findings.

Reviewer 2 ·

Basic reporting

1. The abstract does not provide enough information about the methodology employed in the study (What you did with the GRU model, give a technical description).

2. The paper contains grammatical errors and awkward phrasing

Experimental design

3. In the Related Work section; mention some recent works that are related to deep learning models (LSTM and GRU).

4. In the Proposed Methodology section, you mentioned that; Various pre-processing methods applied to the cryptocurrency dataset prepared it for deep learning processing. Clearly mention all the pre-processing steps and why you taken those particular pre-processing techniques.

5. Ethereum (ETH), Bitcoin (BTC), and Litecoin (LTC), are examined for exchange rate predictions in this study. From where (which repository) you taken the dataset, mention it.

6. Figure 12, labeling Training and Testing loss of ETH but in the figure, it is showing in figure Bitcoin price prediction.

7. Real and predicted price, Training and Testing loss of Litecoin (LTC), mention it clearly it is quite confusing. Give Figure names, numbers, and label it properly.

Validity of the findings

8. In the Materials & Methods section mention all the technical indicators properly for LSTM and GRU.

9. In the Discussion section go properly what is your contributions for these financial predictions.

10. The conclusion does not adequately summarize the key findings of the study.

Annotated reviews are not available for download in order to protect the identity of reviewers who chose to remain anonymous.

---

## Round 0.2 · accepted · Accept

Dear Authors

Thank you for addressing the reviewers' comments. The manuscript now seems ready for publication.

Best wishes,

Reviewer 1 ·

Basic reporting

The authors have sufficiently addressed my concerns in the revised version of the manuscript.

Experimental design

The experimental design and reporting are sufficient

Validity of the findings

The findings seem sufficiently novel, and he reporting is satisfactory

Reviewer 2 ·

Basic reporting

All the comments are addressed by the authors.

Experimental design

The modified manuscript looks good, and it meets the journal standard.

Validity of the findings

Materials & Methods section now includes a detailed description of the technical indicators and hyperparameters used for both LSTM and GRU models, including neuron count, dropout rates, and activation functions.

Additional comments

Overall the modified version is good.